# Assessing the Role of Cannabis Use on Cortical Surface Structure in Adolescents and Young Adults: Exploring Gender and Aerobic Fitness as Potential Moderators

**DOI:** 10.3390/brainsci10020117

**Published:** 2020-02-22

**Authors:** Ryan M. Sullivan, Alexander L. Wallace, Natasha E. Wade, Ann M. Swartz, Krista M. Lisdahl

**Affiliations:** 1Department of Psychology, University of Wisconsin-Milwaukee, Milwaukee, WI 53211, USA; rmsul@uwm.edu (R.M.S.); walla228@uwm.edu (A.L.W.); 2Department of Psychiatry, University of California, San Diego, La Jolla, CA 92093, USA; natashaewade@gmail.com; 3Department of Kinesiology, University of Wisconsin-Milwaukee, Milwaukee, WI 53211, USA; aswartz@uwm.edu

**Keywords:** cannabis, gyrification, surface area, cortical surface structure, aerobic fitness, gender

## Abstract

Cannabis use in adolescents and young adults is linked with aberrant brain structure, although findings to date are inconsistent. We examined whether aerobic fitness moderated the effects of cannabis on cortical surface structure and whether gender may play a moderating role. Seventy-four adolescents and young adults completed three-weeks of monitored abstinence, aerobic fitness testing, and structural magnetic resonance imaging (sMRI). Whole-sample linear regressions examined the effects of gender, VO_2_ max, cannabis use, and their interactions on the surface area (SA) and local gyrification index (LGI). Cannabis use was associated with greater cuneus SA. Gender-by-cannabis predicted precuneus and frontal SA, and precentral, supramarginal, and frontal LGI; female cannabis users demonstrated greater LGI, whereas male cannabis users demonstrated decreased LGI compared to non-users. Aerobic fitness was positively associated with various SA and LGI regions. Cannabis-by-aerobic fitness predicted cuneus SA and occipital LGI. These findings demonstrate that aerobic fitness moderates the impact of cannabis on cortical surface structure, and gender differences are evident. These moderating factors may help explain inconsistencies in the literature and warrant further investigation. Present findings and aerobic fitness literature jointly suggest aerobic intervention may be a low-cost avenue for improving cortical surface structure, although the impact may be gender-specific.

## 1. Introduction

Cannabis is the most used “illicit” substance worldwide with estimated lifetime prevalence rate of 16.9%, with the highest rates in the United States and New Zealand [1]. Specifically within the United States, cannabis is the second most commonly used substance within adolescents and young adults [2]. Approximately 29.7% of adolescents (Grades 8, 10, and 12) [3] and 52% of young adults (aged 18–25) [4] have used cannabis within their lifetime; and, these prevalence rates can vary by state policies [5]. Heavy and chronic cannabis use is associated with adverse psychopathological [6], neurocognitive, and aberrant brain morphology outcomes [7]. Yet, distinct structural changes are less understood [8] and results are not always consistent [7]. Therefore, further research is necessary to elucidate potential moderators of cannabis effects that may explain individual differences.

Cannabis contains cannabinoids, including delta-9-tetrahydrocannabinol (THC) and cannabidiol (CBD). THC is the main psychoactive ingredient and interacts with the endocannabinoid system, affecting the brain through binding to its cannabinoid receptor type 1 (CB1), which is widely distributed throughout the cerebral cortex and principally involved in neuromodulation [9]. Repeated and regular exogenous cannabis exposure can affect CB1 binding [10] and downregulation [11], as well as effects on the brain structure and function (e.g., white matter integrity, functional connectivity, and cerebral blood flow) [12]. Frequent cannabis use, especially high-potency THC products, is associated with structural alterations in medial-temporal, frontal, limbic, and cerebellar regions [7,8,13], however, the extant literature regarding definitive effects on brain structure have garnered mixed findings [14]. Specific investigations into volumetric indices (i.e., cortical thickness and volume) has shown differences between cannabis users and non-users in several areas, including, the hippocampus [15,16], prefrontal cortex [17], right amygdala [18,19], right fusiform [20], orbitofrontal cortex [21,22], inferior parietal cortex [21], anterior cingulate [23,24], precentral gyrus [23,25], superior frontal gyrus [23], right thalamus [25], and cerebellum [26]; see review [27]. Lorenzetti, Chye, Silva, Solowij and Roberts [27] noted further investigation is needed to elucidate potentially mitigating variables in this relationship. Nonetheless, reviews largely center volumetric indices due to a large proportion of literature examining these outcomes. However, less is known on the relationship between cortical surface structure (i.e., cortical gyrification and surface area) and cannabis use and thus is the focus of the present study.

Cortical folding or local gyrification index (LGI) optimizes cortico-connectivity [28]. LGI has been shown to continue development well into adulthood [29] and it is hypothesized to be more susceptible to environmental factors, such as exogenous drug exposure [30,31,32,33]. Cortical surface area (SA) reaches peak levels within adolescence [34] and decreases with age [29], with development largely attributable to heritability [35]. As stated previously, repeated exogenous cannabis exposure can cause alterations in the endocannabinoid system, which plays a role in neuromodulation, pruning, and white matter development. Thus, it is postulated this exposure could disrupt developmental trajectories of LGI and SA, however, this association may be more complex than previously conceptualized. Despite this, only a few studies have examined LGI and SA in cannabis users. Mata, et al. [36] found decreased LGI (i.e., flattening) in the bilateral temporal lobes and left prefrontal cortex within cannabis users; however, no global hemispheric differences in SA were observed. Previous ROI analyses from our lab found decreased LGI in frontal, medial, and ventral medial poles in regular young adult cannabis users, and marginal differences in SA—indicating that cannabis may impact LGI more diffusely compared to SA [37]. Filbey, et al. [38] found no differences in LGI between early and late onset of cannabis users (i.e., age of onset 16.5 and 19 years old, respectively); however, users with earlier onset showed a significant relationship between heavier and longer duration of cannabis use and decreased LGI in prefrontal regions. In a six-year prospective longitudinal study in younger adolescents (aged 12–14 at baseline), decreases in bilateral medial orbitofrontal cortex and right insula SA were found after alcohol and cannabis initiation, though the alcohol-only group demonstrated the most robust findings [39]. Lastly, a recent multi-site analysis in 261 cannabis users and controls found no differences within SA and LGI in regard to cannabis use, cannabis dependence, and age of onset [40]; however, this sample consisted of adult participants (rather than adolescents and young adults) and fine-tuned patterns of use were not ascertained. Furthermore, a majority of these studies were methodologically observational did not examine abstained users to elucidate chronic rather than acute effects. Consequently, the relationship between cannabis and cortical morphometry remains unclear; one potential reason for these inconsistent findings may be moderating factors that put some individuals at a higher risk for negative structural consequences compared to others.

Aerobic exercise (AE) has been linked with positive impacts on the brain. Increasing overall AE in animal models has been linked with increased c-Fos expression [41], brain-derived neurotrophic factors (BDNF) [42], cell proliferation in the hippocampus [43], decreased inflammatory response [44] and oxidative stress [45], and blocking deleterious alcohol-related effects on the hippocampus and dentate gyrus [46,47]. In humans, the interplay between overall aerobic fitness and brain health has been well-established in older adults [48,49,50], but less is known in younger adults or adolescents. Previous research has linked levels of aerobic fitness with better neuropsychological performance in adolescents [51,52] and young adults [53,54], brain structure in children [55], and volumetric differences in young adults [56,57]. Interestingly, depending on levels of aerobic fitness (i.e., high versus low), adolescents display differing functional activation despite similar behavioral performance [58]. Therefore, AE can serve as a potential moderator for brain morphometry in youth. Further, acute AE releases naturally-occurring endocannabinoids [59]; therefore, it is theorized that AE may boost endocannabinoid signaling, potentially counteracting the downregulation effects of exogenous cannabis use [60]. Supporting this hypothesis, our lab recently reported that highly aerobically-fit cannabis users demonstrated better performance in neurocognitive measures compared to users with low aerobic fitness [53]. Further, an AE intervention with sedentary cannabis users found a decrease in craving and use at completion of the intervention [61]. Despite this, few studies explore the relationship between AE and cannabis use on structural brain outcomes.

Another frequent moderating factor in cannabis use research more broadly is gender or sex differences [62,63,64,65,66]. Neuroimaging evidence has pointed towards gender differences in the relationship between chronic cannabis use and structural outcomes [15,17,19,26]; see review [13]. However, none of the aforementioned studies examining LGI or SA outcomes considered gender as a moderator [36,37,38,39,40]. Gender differences are observed within typical assessments of cortical surface structure indices. For example, females exhibit greater cortical complexities (i.e., LGI) overall and in more frontal regions compared to males [67,68], whereas males generally have exhibited larger SA compared to females [69,70,71]. Thus, an investigation into gender differences in cannabis effects is warranted. Finally, gender differences are seen in the effects of AE on cognitive outcomes [72,73]. Consequently, the possibility of gender differences in the impact of AE and cannabis on cortical surface structure is plausible.

The aim of the present study is to look at both aerobic fitness and gender as potentially moderating the relationship between regular cannabis use and cortical surface structure indices, LGI and SA, in adolescents and young adults following three weeks of monitored cannabis abstinence. We expect to see associations between greater aerobic fitness level and increased SA and LGI, and for cannabis users who are aerobically fit to demonstrate fewer cortical abnormalities. We will also examine whether gender moderates the relationship between cannabis and brain morphometry.

## 2. Materials and Methods

### 2.1. Participants

Participants were recruited for a larger parent study through advertisements and flyers in the local community and college. Seventy-four participants in the present study (cannabis users = 36, non-using controls = 38) were between the ages of 16 and 26 years (M = 21.1, SD = 2.6), were generally split for gender (44.6% female), and racial identities consisted of predominantly: Caucasian (64.9%), Asian (10.8%), Multi-racial (10.8%), and African-American (8.1%). (See Table 1).

Participants were included in the parent study if they were right-handed, spoke English, and were willing to abstain from substance use over a three-week period. Exclusion criteria for the parent study included having an independent DSM-IV Axis I (mood, anxiety, psychotic, or attention) disorder, current use of psychoactive medications, major medical or neurological disorders (including metabolic disorders), loss of consciousness >2 min, history of learning disability or intellectual disability, prenatal medical issues or premature birth (gestation <35 weeks), MRI contraindications (pregnancy, claustrophobia, metal in body), reported significant prenatal alcohol exposure (≥4 drinks in a day or ≥6 drinks in a week), prenatal illicit drug exposure, or prenatal nicotine exposure (average > 5 cigarettes per day longer than 1 month), elevated Physical Activity Readiness Questionnaire (PARQ) [74] scores indicating difficulty engaging in VO_2_ max testing, or excessive other illicit drug use (>20 times of lifetime use for each drug category, including cannabis use for non-using control participants). Participants were also balanced at screening for active vs. sedentary physical activity, based on International Physical Activity Questionnaire (IPAQ) [75] score.

The present sample consisted of cannabis users who are categorized as current users who used cannabis at least 44 times in the last year (i.e., nearly weekly) and at least 100 lifetime uses (i.e., nearly weekly for two years). Non-using controls used cannabis no more than 5 times in the past year and less than 20 times in their lifetime [53,66,76,77]. If participants from the parent study (parent study total *N* = 100) did not meet these defined group thresholds, or if participants did not complete VO_2_ max or MRI protocol, they were excluded from the present study (excluded from present study *N* = 26).

### 2.2. Procedures

Data was ascertained from a larger parent study examining the independent and interactive effects of cannabis use and aerobic fitness on neurocognitive outcomes in adolescents and young adults (R01 DA030354; PI: Lisdahl); all aspects of the protocol were approved by the University of Wisconsin-Milwaukee IRB (Study #: PRO00016025). Potential participants who expressed interest in the parent study were asked, over the phone, for demographic information (including age, gender, race/ethnicity, and educational attainment) and screened through an initial semi-structured interview for independent lifetime and past-year Axis I Disorders other than substance use disorder. If determined eligible, study staff obtained a written consent from participants who were aged 18 or older at the start of participation or obtained written assent after parent consent was obtained for minors under 18 years old. Additionally, parents of participants were consented for a parent-administered phone interview that screened for medical, psychiatric and prenatal history before an in-person session was conducted.

Participants who were eligible for the study came in for five study sessions over the course of three weeks. The first three sessions occurred one week apart and consisted of a brief neuropsychological battery (explained in more detail in Wallace, et al. [81]) and urinary drug analysis. Sessions four and five occurred at least one week after session three and consisted of ascertaining body composition, aerobic fitness VO_2_ maximum (VO_2_ max) testing, and then a sMRI that occurred within 24 to 48 h of each other. During the entire study period, participants were asked to abstain from cannabis, alcohol, and other drug use (other than tobacco), which was confirmed through urine, breath, and sweat toxicology screening, which was administered to all participants across all study sessions. If they tested positive for illicit drug use, showed an increase in THCCOOH levels, or had a breath alcohol concentration greater than 0.000 at the start of session two or three (i.e., before VO_2_ max and sMRI procedures), participants were asked to conduct the session after a week of abstinence. Participants were not allowed to complete session four (VO_2_ max) or session five (MRI scan) if they tested positive for any illicit drug use, a rise in THCCOOH levels, or had a breath alcohol concentration greater than 0.000, and instead were allowed to continue their involvement in the study from session one. Participants who used tobacco were asked to abstain from use an hour before the MRI scan.

### 2.3. Measures

#### 2.3.1. Detailed Phone Screen

*Physical Activity*—Extent of physical activity was assessed with the IPAQ [75] and physical ability to engage in VO_2_ max testing was assessed with the PAR-Q [74].

*Lifetime Substance Use Patterns*—Overall patterns of drug and alcohol use were determined by administering the Customary Drinking and Drug Use Record (CDDR) [82] at baseline to measure frequency of cannabis, alcohol, nicotine, and other drug use, SUD symptoms, and the age of onset for first time and regular (i.e., weekly) use.

*Mini Psychiatric Interview*—The Mini International Psychiatric Interview (MINI) [83] or MINI-Kid [84] was administered to participants and parents of minors to screen out for psychiatric comorbidities.

#### 2.3.2. Session Measures

*Past Year Substance Use*—A modified version of the Timeline Follow-Back (TLFB) was conducted by trained research assistants to assess substance use patterns on a week-by-week basis capturing use within the past year, while providing memory cues such as personal events and holidays [66,78]. Substances were measured by standard units [alcohol (standard drinks), nicotine (number of cigarettes and hits of chew/snuff/pipe/cigar/hookah), cannabis (all methods converted to joints or mg in concentrates), ecstasy (number of tablets), sedatives (number of pills or hits of GHB), stimulants (cocaine and methamphetamine use converted to mg and number of amphetamine pills), hallucinogens (number of hits or occasions of ketamine/salvia/shrooms/other hallucinogens), opioids (number of hits of heroin/opium), and inhalants (number of hits)]. Days of cannabis abstinence at scan were calculated from date of last cannabis use based on the TLFB and date of scan.

*Verifying Drug Abstinence*—Participants were expected to remain abstinent from alcohol and other drugs (except tobacco) throughout the course of the study, thus abstinence was evaluated at each session through urine toxicology. The ACCUTEST SplitCup 10 Panel drug test measures amphetamines, barbiturates, benzodiazepines, cocaine, ecstasy, methadone, methamphetamines, opiates, PCP, and THC. Urine samples were also tested using NicAlert to test cotinine level, a metabolite of nicotine. Participants also wore PharmChek Drugs of Abuse Patches, which continuously monitor sweat toxicology for the presence of cocaine, benzoylecgonine, heroin, 6MAM, morphine, codeine, amphetamines, methamphetamine, delta-9-tetrahydrocannabinol (THC), and phencyclidine. Participants additionally underwent breathalyzer screens to test for alcohol use at the start of each session.

*Anthropometric Measures*—Height and weight were measured in light clothes and without shoes. Body Mass Index was calculated as weight divided by height squared (kg/m^2^).

*Body Fat Percentage*—an electrical bioimpedance analysis system was utilized to measure body fat percentage [The Tanita Body Composition Analyzer, TBF-300 (Tanita Corporation, Tokyo, Japan)].

*VO_2_ Maximum*—Participants were instructed to refrain from food and caffeine for 4 hours prior to the exercise test. Prior to each exercise test, the metabolic measurement system, ParvoMedics TrueOne 2400 (ParvoMedics, Salt Lake City, UT, USA) was calibrated using a two-point calibration for the gas analyzers (room air and certified gas: 4.008% CO_2_, 15.98% O_2_, balance N_2_) and a 3 Liter syringe for the pneumotachometer. Participants were fitted with the rubber mouthpiece connected to a Hans Rudolf 2700 series two-way nonrebreathing valve (Kansas City, MO, USA), nose clip, and heart rate strap (Polar Wearlink 31, Kempele, Finland) for the measurement of heart rate and collection of expired gases. Participants completed a maximal incremental exercise test on a treadmill (Full Vision Inc., TMX425C Trackmaster, Newton, KS, USA) following the Bruce Protocol until volitional fatigue. Expired gases were measured continuously using a ParvoMedics TrueOne 2400 metabolic measurement system (ParvoMedics, Salt Lake City, UT, USA). Criteria for determination of attainment of VO_2_ max were based on those recommended [85]. Metabolic data were averaged over 1 min and exported for analysis.

*MRI Acquisition*—Structural MRI scans were acquired on a 3T Signa LX MRI scanner (GE Healthcare, Waukesha, WI, USA) using a 32-channel quadrature transmit/receive head coil. High-resolution anatomical images were acquired using a T1-weighted spoiled gradient-recalled at steady-state (SPGR) pulse sequence (TR = 8.2 ms, TE = 3.4 s, TI = 450 and flip angle of 12°). The in-plane resolution of the anatomical images was 256 × 256 with a square field of view (FOV) of 240 mm. One hundred fifty slices were acquired at 1 mm thickness.

*Processing Pipeline*—Participant structural scans were processed in a standard processing pipeline within FreeSurfer version 5.3 (http://surfer.nmr.mgh.harvard.edu/fswiki/recon-all). T1-weighted 3D anatomical datasets underwent an automated pipeline for motion correction, nonparametric non-uniform intensity normalization, Montreal Neurologic Institute transformation, removal of non-brain materials, skull-stripping, and topology correction. Preprocessed scans were visually examined, and manual edits were made when appropriate. Surface based data created from FreeSurfer’s automated pipeline was utilized for both surface area and LGI analyses. LGI data were computed from pial surface files in accordance with Schaer, et al. [86].

### 2.4. Statistical Analysis

Differences in demographic variables were examined using ANOVAs and Chi-square tests in *R* [87]. A series of multivariate regressions were run with cannabis group, gender, VO_2_ max levels, and their interactions (cannabis group*gender, cannabis group*VO_2_ max, and cannabis group*VO_2_ max*gender) as the independent variables of interest; covariates included past year alcohol use (i.e., total standard drinks) and cotinine levels at the time of VO_2_ max testing. Separate regressions were run measuring for SA [88] and LGI [89]. Analyses were done separately between each hemisphere (right and left) and smoothed with a global Gaussian blur at FWHM of 10 for all analyses. Corrections for multiple comparisons were made using Monte Carlo simulations at a cluster wise probability (*cwp*) of *p* = 0.05, correcting across both hemispheric spaces, and smoothed at FWHM of 20 for SA and 25 for LGI corrections. Decisions about statistical significance were made at *p* < 0.01 for all analyses. Regions that meet statistical significance were annotated using the Desikan-Killiany Atlas [90].

## 3. Results

### 3.1. Demographic Data

There were no significant differences between cannabis user and non-user groups in age [*F*(1,72) = 1.22, *p* = 0.27], gender [*χ*^2^(1) = 1.43, *p* = 0.23], race [*χ*^2^(6) = 5.87, *p* = 0.44], ethnicity [*χ*^2^(2) = 2.69, *p* = 0.26], educational attainment [*F*(1,72) = 0.08, *p* = 0.78], VO_2_ max [*F*(1,72) = 1.10, *p* = 0.30], and body fat percentage [*F*(1,72) = 1.27, *p* = 0.27]. There was a significant difference in amount of lifetime [*F*(1,72) = 28.5, *p* < 0.001] and past year [*F*(1,72) = 35.9, *p* < 0.001] cannabis use, alcohol consumed within the past year [*F*(1,72) = 17.6, *p* < 0.001], past year tobacco use [*F*(1,72) = 7.46, *p* = 0.008], and cotinine levels at VO_2_ max testing [*F*(1,72) = 9.24, *p* = 0.003]; past year alcohol use and cotinine levels were included as covariates in all analyses.

There were no significant differences between males and females for age [*F*(1,72) = 0.05, *p* = 0.83], race [*χ*^2^(6) = 8.23, *p* = 0.22], ethnicity [*χ*^2^(2) = 1.48, *p* = 0.48], educational attainment [*F*(1,72) = 0.18, *p* = 0.67], past year tobacco use [*F*(1,72) = 3.82, *p* = 0.06], cotinine levels at VO_2_ max testing [*F*(1,72) = 2.54, *p* = 0.12], or past year [*F*(1,72) = 3.56, *p* = 0.06] and lifetime [*F*(1,72) = 3.57, *p* = 0.06] cannabis use. Significant differences between genders were observed for past year alcohol use [*F*(1,72) = 4.29, *p* = 0.04], VO_2_ max [*F*(1,72) = 52.62, *p* < 0.001], and body fat percentage [*F*(1,72) = 60.9, *p* < 0.001].

Within cannabis users, there was no significant differences between genders in past year alcohol use [*F*(1,34) = 1.01, *p* = 0.32], past year tobacco use [*F*(1,34) = 2.69, *p* = 0.11], cotinine levels at VO_2_ max testing [*F*(1,34) = 1.59, *p* = 0.22], past year [*F*(1,34) = 1.72, *p* = 0.20] or lifetime [*F*(1,34) = 1.83, *p* = 0.19] cannabis use, age of first regular cannabis use onset [*F*(1,34) = 0.37, *p* = 0.55], or days of cannabis abstinence prior to sMRI [*F*(1,34) = 1.26, *p* = 0.27].

### 3.2. Primary Analyses

Whole-sample analyses were conducted examining the effects of cannabis, VO_2_ max, gender and their interactions on SA and LGI, while covarying for past year alcohol use and cotinine level at time of VO_2_ max testing. (See Table 2 and Table 3).

#### 3.2.1. Cannabis Results

*Surface area.* Cannabis users demonstrated significantly larger SA in the left cuneus compared to controls [*t*(58) = 2.64, *cwp* = 0.006] (See Figure 1). *Gyrification.* There was no main effect of cannabis group observed for LGI.

#### 3.2.2. Cannabis*Gender

*Surface area.* There was a significant interaction between cannabis group and gender in the left precuneus [*t*(58) = −3.31, *cwp* = 0.006], left rostral middle frontal [*t*(58) = −2.30, *cwp* = 0.0006], and two right superior frontal [*t*(58) = −3.49, *cwp* = 0.003; *t*(58) = −2.25, *cwp* = 0.002] regions for SA. Female cannabis users had increased SA in these regions compared to non-using females, whereas male cannabis users demonstrated decreased SA compared to non-using males, except in the second right rostral middle region, where cannabis using males and females exhibited less SA compared to non-using males and females. *Gyrification.* There was a significant interaction between cannabis group and gender in the left precentral [*t*(58) = −2.89, *cwp* = 0.0001], left lateral orbitofrontal [*t*(58) = −2.53, *cwp* = 0.0004], and right supramarginal [*t*(58) = −3.78, *cwp* = 0.0001] regions for LGI. Female cannabis users had increased LGI in these regions compared to non-using females and male cannabis users had decreased LGI compared to non-using males; except in left lateral orbitofrontal region, where female cannabis users displayed slightly less gyrification compared to non-using females and male cannabis users had decreased LGI compared to non-using males. (See Figure 2).

#### 3.2.3. VO_2_ Results

*Surface area.* In both the cannabis users and non-users, significant relationships between greater VO_2_ max and larger SA were found in the left superior parietal [*t*(58) = 4.65, *cwp* = 0.007], left inferior parietal [*t*(58) = 4.24, *cwp* = 0.0001], right inferior parietal [*t*(58) = 3.89, *cwp* = 0.0001], and right inferior temporal [*t*(58) = 3.27, *cwp* = 0.0001] regions. *Gyrification.* Participants displayed a significant relationship between greater VO_2_ max and greater LGI in the left superior temporal [*t*(58) = 5.17, *cwp* = 0.0001], right lateral orbitofrontal [*t*(58) = 3.27, *cwp* = 0.0001], and right inferior parietal [*t*(58) = 2.78, *cwp* = 0.0015] regions. (See Figure 3).

#### 3.2.4. Cannabis*VO_2_

*Surface area.* There was a significant interaction between VO_2_ max and cannabis group in the left cuneus [*t*(58) = −3.72, *cwp* = 0.0001] region; non-using controls demonstrated a strong positive relationship between VO_2_ max and increased SA, whereas cannabis users demonstrated a negative relationship. *Gyrification.* There was a significant interaction between VO_2_ max and cannabis group in the left lateral occipital region [*t*(58) = −3.71, *cwp* = 0.0004]; non-using controls demonstrated a positive relationship between VO_2_ max and increased LGI, whereas no trend was observed for cannabis users. (See Figure 4).

#### 3.2.5. Cannabis*VO_2_*Gender

In order to further characterize potential gender differences, exploratory three-way interaction analyses were investigated. Results revealed an interaction between group, gender, and VO_2_ was shown for LGI in the right supramarginal region [*t*(58) = 2.57, *cwp* = 0.009]. Representing a potential area of further investigation with larger sample sizes.

## 4. Discussion

Worldwide, cannabis use rates are known to vary across countries [1] and age of onset is typically between 18 to 19 years old [91]. Additionally, given the ongoing policy debate surrounding cannabis in the United States, its long-term effects in youth are of increasing scientific and clinical interest. To date, findings regarding the impact of regular use on brain morphometric outcomes have not been entirely consistent [12]. Reasons underlying these differential findings include demographic differences in samples, especially regarding gender, or other potential resilience or risk factors including extent of aerobic fitness [53,60,92]. Here, we attempt to clarify the confusion regarding cannabis effects by examining two potential moderators: aerobic fitness and gender. We found that after three weeks of monitored drug abstinence, cannabis users demonstrated greater SA in the cuneus. Notably, we also found significant interactions between gender, aerobic fitness, and cannabis use on SA and LGI outcomes in prefrontal and parietal cortical regions. Generally speaking, males appeared more sensitive to cannabis impacts following abstinence, and further, participants demonstrated a positive relationship between aerobic fitness and cortical surface structure more broadly.

After accounting for gender, alcohol, cotinine, and aerobic fitness level, the whole-group findings demonstrated increased SA in the cannabis users compared to controls in a region denoted as the cuneus in the left hemisphere. The cuneus is functionally connected with parietal and other occipital regions for the purposes of integrating visual information [93] and contains CB1 receptors [94]. This region has a non-linear trajectory of neuronal maturation [95] and thus larger SA in cannabis users may be evident of delayed development in this region compared to non-using counterparts. Further, cannabis users have previously demonstrated abnormal dose–response blood-oxygenated-level-dependent signaling in the cuneus [96] along with aberrant functional activation in the occipital region more broadly [13,97]. The present analysis builds on this literature by demonstrating SA in this region is structurally different between adolescent and young adult cannabis users and non-users. Yet, this is inconsistent with prior findings demonstrating either no differences between cannabis users and controls in cuneus SA [36,38,40], or research reporting decreased LGI in prefrontal and temporal lobes in cannabis users [36,37] and early onset cannabis use [38], and reduced SA in comorbid cannabis and alcohol-using adolescents [39]. These inconsistencies may be due to differences in gender distribution across the studies, all of which skewed male [55% (current study), 56% [39], 64% [37], 67% [40], 73% [38], and 77% [36] male], varying levels of aerobic fitness in the sample (current sample was balanced for recent physical activity levels), or accounting for aerobic fitness levels within the statistical design. In addition, these inconsistencies may be due to differing age of samples [36,38], with younger samples undergoing greater neurodevelopment; or, due to analysis design—e.g., whole-brain analyses compared to ROI analyses [37]. None of the prior studies examining these outcomes tested whether gender or aerobic fitness moderated these effects or controlled for aerobic fitness level. Although we found the relationship between cannabis use and cortical surface structure was moderated by two-way interactions which were observed between gender, cannabis and aerobic fitness in frontal, cingulate, and parietal regions; regions that have been found to be abnormal in previous studies [21,23,24,36,37,39]. Thus, we will focus on these novel interactions.

Male cannabis users had lower SA in left precuneus, rostral middle frontal, and right superior frontal and lower LGI in left precentral, lateral orbitofrontal, and right supramarginal regions compared to non-using males—after accounting for alcohol use, cotinine level, and aerobic fitness level. Of note, our sample underwent three-weeks of monitored abstinence from cannabis and other drugs of abuse before structural scans were conducted; thus, THC and other exogenous cannabinoids were metabolized out of the system [98], representing chronic rather than residual associations. These fronto-parietal cortical findings are consistent with several studies reporting abnormal brain morphometry in cannabis users, including reduced volume [21,23,24,99] and lesser cortical gyrification [36,37] in samples that were primarily male. These overall interactions did uncover SA abnormalities in the male cannabis users, which is inconsistent with prior null or marginal SA findings [37,40,100], however, future gender-stratified analyses are needed to ascertain gender-specific mechanisms in the relationship between cortical surface structure and cannabis. In females, cannabis use was linked with greater SA in left precuneus, rostral middle frontal, and right superior frontal and greater LGI in left precentral and supramarginal regions compared to non-using females. Interestingly, female cannabis users also exhibited slightly reduced SA in another right superior frontal region and within left lateral orbitofrontal LGI. The increased SA and LGI findings could suggest that cannabis use in adolescent and young adult females is either advantageous, or potentially represents delayed pruning, as the female trajectory of pruning has an earlier rate compared to males [101]. Of note, our groups did not differ in age and we are capturing a cross-sectional snapshot potentially depicting a delay due to cannabis use specifically in females, whereas, smaller volumes in male cannabis users could be interpreted as detrimental effects associated with use. Interpretation of the present female findings are consistent with prior studies reporting greater brain volumes in female adolescent cannabis users [17,19]. More pronounced differences in males is consistent with prior findings that male cannabis users were more vulnerable to neurocognitive deficits in sequencing ability compared to females [66]. Interestingly, across genders, LGI differences were equally apparent compared to SA differences; indicating that LGI, an understudied structural index in addiction literature, is potentially susceptible to environmental influences [30,31,32,33] and is a viable avenue for further investigation. It is notable that our prior work has found cognitive deficits in psychomotor speed, working memory, sustained attention, and inhibitory control in an overlapping sample of both male and female young adult cannabis users [53,66,76], findings that are consistent with recent meta-analysis [102] and longitudinal [103] studies. Thus, structural deviancies may represent a mechanism for downstream cognitive functioning, although future studies are needed to assess whether structural changes directly impact function.

These differential gender patterns in cannabis findings may be due to multiple reasons. One potential underlying cause is differential substance use patterns. While male or female cannabis-using groups did not statistically differ from one another in their use patterns, our male users had more cannabis use on average compared to female users. This is consistent with previous literature indicating males have more severe use patterns compared to females [63]. Moreover, male users may also be more prone to using greater individual doses or more potent THC products [104], although the current study cannot address that possibility. Male sensitivity may also be due to differences in CB1 receptor density, as greater density in males is observed in animal models [105,106]. These gender differences may also be influenced in part by inherent sexual dimorphism in neurodevelopment [107,108] and the introduction of cannabis into these staggered developmental trajectories. Another possibility is that there was less power to detect differences in females, as their sample size was smaller overall (female: *n* = 33 vs. male: *n* = 41) and within cannabis users (female: *n* = 13 vs. male: *n* = 23); thus, additional research utilizing larger samples specifically examining the relationship between gender and cannabis use on these morphometric outcomes is needed.

Regarding aerobic fitness, results demonstrated several main effects of VO_2_ max across both cortical surface structure indices in both the cannabis users and non-users. Greater VO_2_ max was associated with increased SA in left superior and inferior parietal, and right inferior parietal and inferior temporal regions; and, increased LGI in left superior temporal, and right lateral orbitofrontal and inferior parietal regions. These findings indicate that in both cannabis users and non-users, VO_2_ max has a strong positive relationship with cortical surface structural indices. This is consistent with previous analyses in our lab showcasing a positive link between AE and neurocognitive outcomes [53], and recent reviews of brain morphological outcomes in aerobic fitness literature [109]. Further, these findings suggest that, similar to non-users, cannabis users appear to have a positive link between aerobic fitness and brain structure in several regions of interest. When examining the interaction between cannabis and aerobic fitness, an interaction was observed in left cuneus SA and left lateral occipital LGI, with non-using controls exhibiting positive associations between increased indices with greater VO_2_ max compared to cannabis users who demonstrated either a flat or negative relationship. This finding suggests that the impact of aerobic fitness may be less pronounced in regular cannabis users in these particular regions. Albeit, main effects of VO_2_ max suggest that both groups generally had a positive relationship between aerobic fitness and brain morphometry; which is consistent with prior studies in adolescents, young adults, and older adults [51,110,111]. On balance, we previously reported that aerobic fitness was linked with superior visual memory, verbal fluency, and sequencing ability and highly fit cannabis users performed better on psychomotor speed, visual memory and sequencing ability compared to low-fit users [53]. It is important to note that our participants had no comorbid metabolic conditions (e.g., hyperlipidemia, hypertension, diabetes) suggesting benefits of aerobic fitness even in physically healthy youth. Possible mechanisms supporting positive effects of aerobic fitness and brain structure are likely multi-factorial. As aforementioned, engaging in AE releases BDNF [42], vascular growth factors [112], insulin-like growth factor-1 (IGF-1) [113], neurogenesis [114], improved catecholaminergic signaling [115], increased c-FOS expression [41], is linked with increased hippocampal volume [116], and reduces inflammation and oxidative stress [44,45]. Future studies are needed to tease apart these potential underlying mechanisms.

As we have postulated previously [53,60], AE intervention may be a plausible avenue to explore in further studies aiming to reduce or ameliorate neurocognitive deviances associated with repeated and chronic cannabis use [92]. Indeed, other groups have reported that aerobically-fit cannabis users reported reduced craving and fewer symptoms of cannabis use disorder compared to unfit users [61,117]. One potential explanation of the interactive association between cannabis use and AE could be more aerobically-fit users are metabolizing exogenous cannabinoids out of the body faster, thus, mechanistically dampening the impact on cortical surface structure integrity and neurocognition more broadly. Previous literature has shown mixed findings in acute AE increasing cannabinoid metabolites (i.e., THCCOOH) [118,119]. Further, AE releases endocannabinoids [120,121,122], which may help mitigate the negative impact of repeated exogenous cannabis exposure. Notable for future studies, it is hypothesized this relationship may be gender-specific; thus, research examining the potential use of aerobic interventions in cannabis users should prioritize investigation of potential gender differences.

There are potential weaknesses to note. First, causality cannot be determined from the present sample, sMRI scans were obtained after regular cannabis use was established. Second, the sample size for female cannabis users was relatively small compared to male cannabis users; still, findings support gender-specific cannabis and aerobic fitness associations through the interactions uncovered. This lends additional evidence for the need to examine sex as a potential moderator of cannabis effects, which has largely been understudied in cortical surface structure indices. Findings may not generalize to other samples of cannabis users with substantially different use patterns, e.g., length of abstinence, age of regular use onset, or other substance use. Third, concerns have been raised within the literature regarding divergent findings based on different brain morphometry detection algorithms and software [123]. While several validation studies of Freesurfer’s surface indices have been published [89,124,125], future work should examine how differing surface-based analyses may produce slightly different regions of interest from cannabis use in adolescents and young adults. Fourth, the relationship between cannabis use and cortical surface structure has other potential moderators of interest; two important ones are genetic factors [126,127,128] and psychiatric comorbidities [129,130,131]. The ability to simultaneously examine several potential moderators of cannabis effects on neurocognitive outcomes will soon be available with the large-scale, prospective, longitudinal Adolescent Brain Cognitive Development (ABCD) Study [132] (www.abcdstudy.org/), which has enrolled over 11,800 youth. Lastly, our average VO_2_ max was lower than age-based norms [79], so assessing a more aerobically fit group may demonstrate even stronger associations with cortical surface structure.

The current study found that cannabis users had larger cuneus SA and male cannabis users exhibited smaller SA and less complex LGI in frontal, cingulate and parietal regions, even after three weeks of monitored abstinence, compared to male non-users. In contrast, female cannabis users generally demonstrated increased SA and LGI in the aforementioned regions. Prospective, longitudinal studies, such as the ABCD Study, are needed to address whether abnormalities in LGI were caused by cannabis or due to premorbid factors, and identifying the potentially gender-specific developmental trajectories in the impact of cannabis use on the brain. We also found that both cannabis users and non-using controls had a significant link between increased aerobic fitness and more complex LGI and larger SA in frontal, parietal, and temporal regions. These findings, combined with our prior report of superior cognitive functioning in aerobically-fit cannabis users [53], support the notion that it is viable to investigate whether enhancing aerobic fitness, through AE, may be a feasible prevention or ameliorative tool aimed at reducing the impact of chronic cannabis on neurocognitive outcomes in adolescents and young adults.

## Figures and Tables

**Figure 1 brainsci-10-00117-f001:**
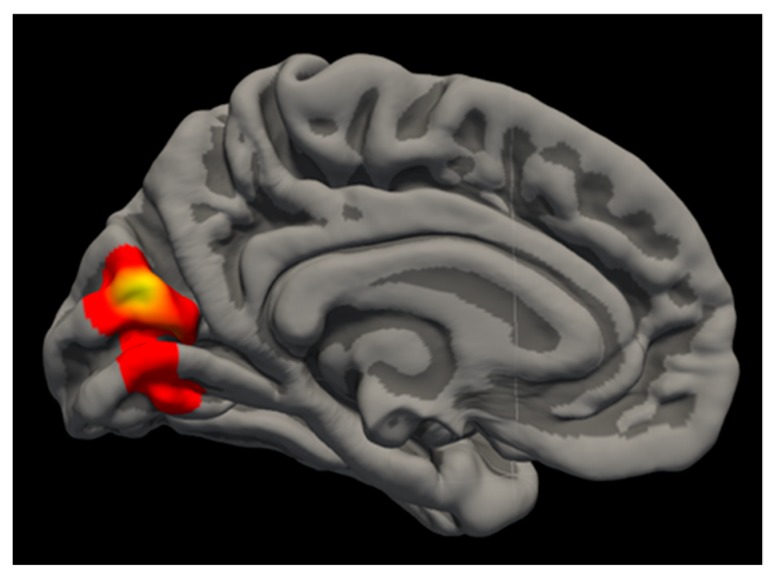
Cannabis Findings. Medial view of cannabis finding within left cuneus SA, with larger area in cannabis users.

**Figure 2 brainsci-10-00117-f002:**
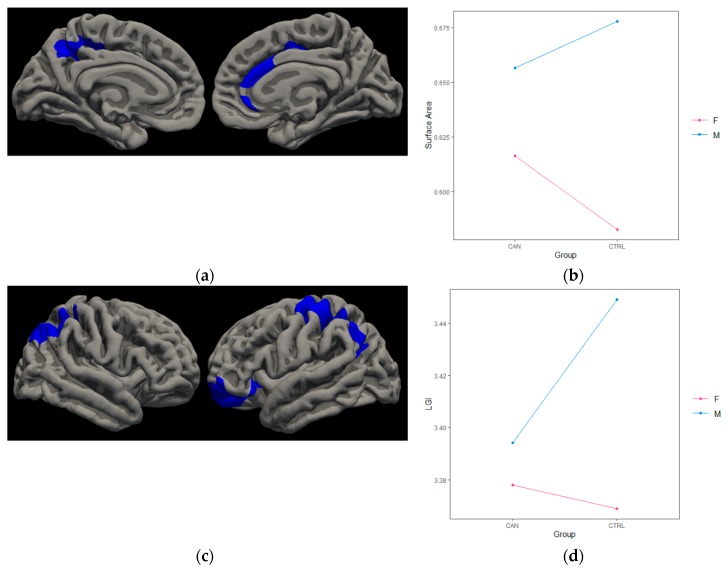
Cannabis*Gender Findings. (**a**) Medial view of significant interaction between cannabis group and gender in left cuneus, left rostral middle frontal (not pictured), and right superior frontal SA. (**b**) Generally, female cannabis users demonstrated more SA in these regions compared to non-using females, whereas male cannabis users demonstrated less SA compared to non-using males (left rostral middle frontal SA finding depicted). (**c**) Lateral view of significant interaction between cannabis group and gender in right supramarginal, left precentral, and left lateral orbitofrontal LGI. (**d**) Generally, female cannabis users had more LGI in this region compared to non-using females, whereas male cannabis users had less LGI compared to non-using males (left precentral LGI finding depicted).

**Figure 3 brainsci-10-00117-f003:**
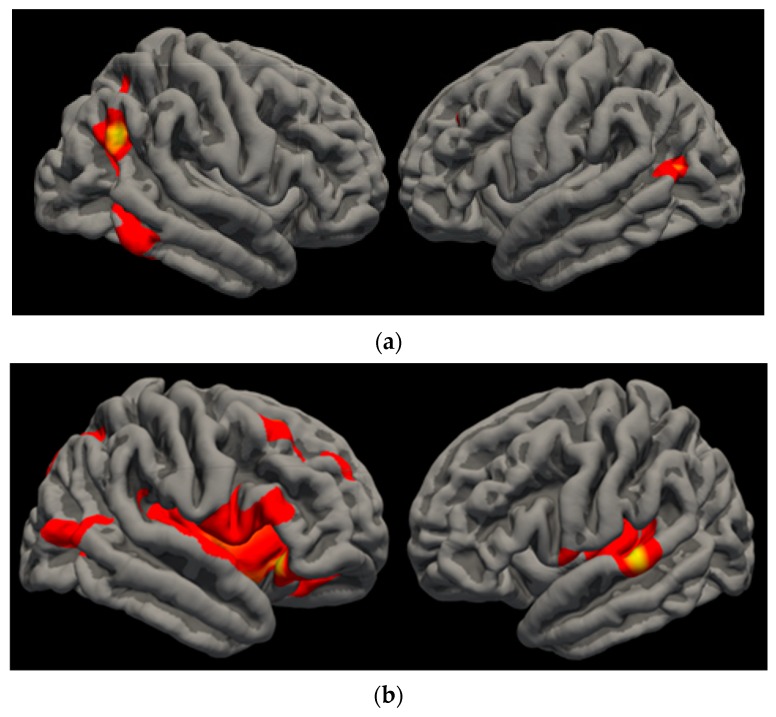
VO_2_ Findings. (**a**) Lateral view of VO_2_ finding observed in right inferior parietal, right inferior temporal, left superior parietal (not pictured), and left inferior parietal SA. Greater VO_2_ was associated with more area in these regions. (**b**) Lateral view of VO_2_ finding observed in right lateral orbitofrontal, right inferior parietal, and left superior temporal LGI. Greater VO_2_ was associated with more gyrification in these regions.

**Figure 4 brainsci-10-00117-f004:**
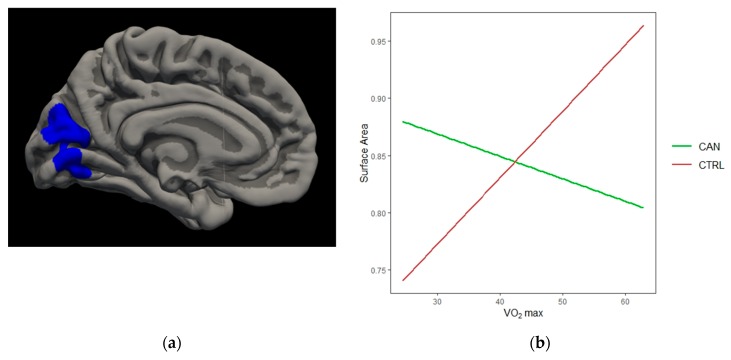
Cannabis*VO_2_ Findings. (**a**) Medial view of significant interaction between VO_2_ and cannabis group in left cuneus SA; (**b**) non-using controls demonstrated a positive relationship between VO_2_ and increased area, whereas cannabis users demonstrated a negative relationship. (**c**) Lateral view of significant interaction between VO_2_ and cannabis group in left lateral occipital LGI; (**d**) non-using controls demonstrated a positive relationship between VO_2_ and increased LGI, whereas no trend was observed for cannabis users.

**Table 1 brainsci-10-00117-t001:** Demographics.

	Cannabis Users			Non-Using Controls		
	All	Male	Female	All	Male	Female
*N*	36	23	13	38	18	20
*M* (SD) or %						
Age (years)	21.4 (2.3)	21.4 (2.4)	21.4 (2.0)	20.8 (2.8)	20.5 (3.1)	21.0 (2.6)
Race (% Caucasian)	58.3%	65.2%	46.2%	71.1%	72.2%	70.0%
Ethnicity (% Non-Hisp)	77.8%	78.3%	76.9%	86.8%	94.4%	80.0%
Educational Attainment (years)	14.0 (1.6)	13.9 (1.8)	14.1 (1.3)	14.1 (2.4)	14.0 (2.9)	14.2 (1.9)
Past yr Alcohol Use ^a,^*	338.7 (300.8)	376.6 (306.2)	271.6 (290.5)	100.6 (173.6)	141.8 (225.1)	63.5 (101.5)
Past yr Tobacco Use ^a,^*	214.6 (483.7)	311.8 (585.1)	42.8 (68.1)	0.5 (1.97)	0.2 (0.43)	0.7 (2.7)
Cotinine Level ^b,^*	2.0 (1.8)	2.3 (2.1)	1.5 (1.0)	1.1 (0.6)	1.1 (0.6)	1.1 (0.6)
Past yr Cannabis Use ^a,^*	428.2 (440.4)	499.9 (510.7)	301.5 (245.4)	0.36 (1.2)	0.7 (1.6)	0.1 (0.22)
Lifetime Cannabis Use ^a,^*	1189.6 (1372.3)	1419.7 (1621.6)	782.5 (625.0)	1.5 (2.9)	1.2 (2.3)	1.8 (3.5)
Age at Regular Cannabis Use Onset (years)	17.5 (1.7)	17.4 (1.9)	17.8 (1.3)	− ^c^	− ^c^	− ^c^
Cannabis Abstinence Length (days) ^d^	31.1 (22.9)	34.3 (27.9)	25.5 (6.5)	− ^c^	− ^c^	− ^c^
VO_2_ maximum ^e,^^	43.7 (9.0)	47.9 (6.6)	36.1 (7.7)	41.4 (9.8)	47.9 (8.8)	35.5 (6.3)
VO_2_ maximum (%ile) ^f^	− ^c^	69.4%	37.2%	− ^c^	68.9%	33.9%
Body Fat (%) ^g,^^	19.1% (8.5)	15.6% (6.9)	25.3% (7.7)	21.6% (10.0)	13.6% (6.1)	28.7% (7.0)

* *p* < 0.001 between cannabis users and non-using controls. ^ *p* < 0.001 between males and females. ^a^ Measured in standard uses on TLFB [78]. ^b^ Measured at VO_2_ maximum testing session. ^c^ Not applicable. ^d^ Calculated from TLFB last cannabis use date and date of sMRI. ^e^ Raw values. ^f^ Gender-specific percentiles were calculated with mean age, using ASCM norms [79,80]. ^g^ Body Fat was ascertained in the same session at which VO_2_ maximum testing was conducted.

**Table 2 brainsci-10-00117-t002:** Surface Area findings.

	*t*	Size (mm^2^)	x	y	z	*cwp*
**Cannabis**						
Left Cuneus	2.639	1706.64	−4.1	−78.6	19.1	0.006
**Cannabis*Gender**						
Left Precuneus	−3.306	1718.12	−10.3	−54.6	46.7	0.006
Left Rostral Middle Frontal	−2.299	2348.85	−44.7	27	31.4	0.0006
Right Superior Frontal	−3.491	1819.72	11.5	9	36.9	0.003
Right Superior Frontal	−2.248	2007.88	23.1	0.4	61.6	0.002
**VO_2_**						
Left Superior Parietal	4.654	1673.98	−28.4	−64.4	39.9	0.007
Left Inferior Parietal	4.236	2535.22	−45.1	−63.9	10	0.0001
Right Inferior Parietal	3.894	3235.39	47.3	−59.7	29.4	0.0001
Right Inferior Temporal	3.268	2877.28	51.4	−55.2	−15.1	0.0001
**Cannabis*VO_2_**						
Left Cuneus	−3.724	2736.75	−4.4	−77.1	21.6	0.0001

**Table 3 brainsci-10-00117-t003:** Gyrification findings.

	*t*	Size (mm^2^)	x	y	z	*cwp*
**Cannabis*Gender**						
Left Precentral	−2.894	4993.5	−37.6	−12.2	62.1	0.0001
Left Lateral Orbitofrontal	−2.533	3240.22	−18.8	51.8	−13.8	0.0004
Right Supramarginal	−3.784	4763.28	49.6	−41.1	40.9	0.0001
**VO_2_**						
Left Superior Temporal	5.174	10682.85	−64.5	−25.2	4.2	0.0001
Right Lateral Orbitofrontal	3.272	13062.8	39	27.3	−9	0.0001
Right Inferior Parietal	2.78	2718.55	34.2	−73.4	37.6	0.0015
**Cannabis*VO_2_**						
Left Lateral Occipital	−3.712	3297.29	−28.2	−95.1	−12.7	0.0004
**Cannabis*VO_2_*Gender**						
Right Supramarginal	2.572	2207.65	48.8	−40.5	40.3	0.009

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
