# Peer review of "Assessing the Role of Cannabis Use on Cortical Surface Structure in Adolescents and Young Adults: Exploring Gender and Aerobic Fitness as Potential Moderators"

_brainsci, 2020, doi:10.3390/brainsci10020117_

Round 1

Reviewer 1 Report

The authors have addressed my comments. I'd only suggest they include the n of the parent study in the revised ms. I'd also suggest one last read through for editing. 

Reviewer 2 Report

Sullivan and colleagues investigated differences in two cortical surface measures between cannabis-using and non-using adolescents and young adults. The authors found differences in surface area and local gyrification index between cannabis-users and controls, as well as interactions with gender and aerobic fitness. A major finding was female cannabis users having largely increased surface area and local gyrification indices compared with controls, whereas male cannabis users largely showed decreased surface area and gyrification indices. The research is original, well-written and particularly relevant in the light of global changes in cannabis legislation.

I have suggestions for major and minor revisions that should be addressed by the authors:

A major concern is the small sample size of cannabis using females (n=13), compared with cannabis using males (n= 23) and non-cannabis using females (n= 20). The author relativises this well with the phrasing ‘exploring gender as moderator’, however, I am concerned that opposing findings for male versus female cannabis-users may be based on imbalances in power. I understand that using a sub-sample of another study renders matching difficult but if collecting additional cannabis using females is not feasible, the imbalance should be made explicit in the methods. For example, line 126 states that participants were gender-balanced (44.6%). This is true across for the whole sample, however, as overall gender comparison is not a major interest, this statement is misleading. I am missing discussion of implications for the reported findings. For example, the function of the cuneus could be elaborated on and what smaller SA in males and larger SA in females could imply. Line 143, a justification is needed as to why inclusion criteria for cannabis group was limited to smoking ‘nearly weekly’. If adolescents rarely smoke more frequently, a reference should be given. ‘Nearly weekly’ cannabis use for a minimum of two years is unlikely to have caused changes in cortical surface structure, begging the question of causality. If there is evidence to suggest otherwise, please elaborate. If no evidence exists, the presented results should to some degree be discussed in the light of motivating cannabis/substance use.

Minor concerns:

Line 17: spell out or define sMRI and define again in the main body of the text. The motivation for asking participants to abstain from cannabis for three weeks needs to be explained in the introduction. In the first two sentences of the introduction: Add information to make more relevant for international readers. Line 41, exchange the semicolon for a period, the sentence is too long. Line 167: Exchange ‘participants’ with ‘participation’. Line 175: I would use ‘sMRI’ as you do in the rest of the text. Line 176: I would remove ‘with none occurring on the same day.’. It is a bit confusing and already implicated by ’24-48 hours’. Line 178: As you write ‘breath, and/or sweat toxicology screening’, what determined whether breath or toxicology screening was done? Line 180: You write ‘any’ subsequent session after baseline but if I understand this correctly, the procedure only refers to week 1-3. If drug use is detected in week 4-5, participants start at week 1. Make this explicit as it is confusing when the reader gets to line 184. Line 199: spell out ‘research assistants’ Line 199: The phrasing might suggest that the research assistants recorded the TLFB weekly over the past year. Clarify that they recorded averages over one week. Linen 213: You report that all participants wore PharmCheck patches but in line 178, it sounds like not every participant is drug screened. Clarify. Line 219: I think it should be (kg/m2 ) 241: I am not familiar with this type on analysis, but did you process everything in a standard processing pipeline or was this just the pre-processing? Line 243: ‘Automatic’ should be ‘automated’ as below. Line 293: Exchange ‘cannabis finding’ for a more meaningful description. On this note, I would generally exchange ‘finding’ with ‘results’ or just remove it where possible, e.g. in Table 2. Cannabis*Gender Finding and below, ‘Finding’ can be removed. Line 307: Exchange ‘Whereas’ for ‘and’ as the statement is not contradicting. Line 354: As in introduction, add information on other countries to make this more relevant to an international readership. Line 364: ‘while all participants demonstrated a positive relationship between aerobic fitness and 364 cortical surface structure.’ Section 3.2.4. suggests that cannabis users did not demonstrate this positive relationship. Line 377: The sentence starting in this line is long and complicated, split for ease of reading. Line 387: Explain this rationale for cannabis abstinence in the introduction, along with a reference for how long residual associations could be a factor. Line 468: I like how you are phrasing the gender differences in a somewhat more moderate way. I think regarding the difference in sample size from female to male cannabis users and female cannabis user to female controls, this is the way to go. Line 475: remove ‘in’

Round 2

Reviewer 2 Report

Dear Mr. Sullivan and colleagues, 

thank you for addressing my comments. I recommend your article to be accepted.

This manuscript is a resubmission of an earlier submission. The following is a list of the peer review reports and author responses from that submission.

Round 1

Reviewer 1 Report

This is an interesting study investigating the relationship between cannabis use and cortical surface structure, and the modifying effects of gender and aerobic fitness. Overall, the manuscript is very well written and easy to follow. I have some comments which I feel should be addressed:

Can the authors please indicate how the current sample differed from the sample in the parent study and how inclusion into the current study was determined. What proportion of the participants from the parent study are included here?

Session 5 (MRI scan) occurred within 24-48 hours after session 4 (VO2 max testing). Can the authors include the minimum time between these two sessions? I’m assuming the two sessions didn’t occur on the same day for any participants?

From my reading of the manuscript, abstinence from alcohol could only really be verified within the last 12-24 hrs if only based on breathalyzer screens.

I would also like to see the rates of tobacco use compared between the two groups and I think tobacco use should be controlled for in all analyses. I also feel that frequency/quantity, or some indication of cannabis involvement, should also be controlled for given the sex differences in use noted in the discussion.

‘marginal differences’ (line 267) - either there were statistically significant differences or there were not.

Line 389 mentions that the main cannabis analysis controlled for a list of variables. This wasn’t clear in the methods. Continuing in that paragraph, the authors indicate that the increased SA/LGI findings are in contrast to previous studies which didn’t control for these variables. This could be investigated here by running the model without these control variables.

The gender stratified analyses include 13 female cannabis users – a fairly large limitation, especially given that the discussion then focuses primarily on the stratified analyses. I suggest more weight is given to the whole sample analyses and the stratified analyses be presented as tentative.

From my reading, the effects of AE on the brain were not apparent in cannabis users. I’m not sure whether the paragraph beginning on line 469 reflects this finding? I also need some convincing that the conclusion on line 505 is supported by the data.

Thank you for the opportunity to review a thoughtful and interesting paper.

Reviewer 2 Report

This manuscript presents interesting data on potential moderators of associations between cannabis use and brain structural findings. The authors use analyses of cortical surface structure to examine whether gender and a measure of aerobic fitness moderate these relationships. The authors report significant interactions between cannabis and gender and cannabis and aerobic fitness in a few brain regions. They interpret the findings to indicate that these may be important factors affecting the inconsistencies in this literature. If properly designed, these findings could be could be influential in our understanding of potential gender differences in cannabis use and its functional consequences. However, I have significant concerns with several aspects of this manuscript, which I think hamper its potential impact. The authors have noted some primary limitations in their manuscript, although I am concerned that they do not place enough emphasis on these limitations in conducting their analyses and interpreting their results. 

Major:

The most significant limitation in the manuscript is statistical power and the type and number of analyses the authors have chosen to conduct. Sufficient power is critical for ensuring that important significant effects are not undetected, but it is also equally important in decreasing chances of finding a false positive. As such, power may drive failures to replicate findings within psychology and neuroscience (Fraley & Vazire, 2014). Since the literature is so variable on morphometry differences associated with cannabis use, as the authors note, the effects they are trying to detect are likely small. With their sample sizes, the authors are drastically underpowered for detection of three-way interactions with even medium effect sizes, and are likely quite underpowered for the detection of two-way interactions with small effect sizes. See blogposts from Roger Giner-Sorolla https://approachingblog.wordpress.com/2018/01/24/powering-your-interaction-2/ and Andrew Gelman: https://statmodeling.stat.columbia.edu/2018/03/15/need-16-times-sample-size-estimate-interaction-estimate-main-effect/. They also perform an excessive number of analyses, and it is not clear whether the Type I error correction they use applies across the whole set of analyses, or just sets (e.g., SA analyses, stratified analyses, etc.). Given these limitations, it is more likely than not that these findings would not replicate. As a related critique, there is only one significant cannabis by gender interaction that shows up in the primary analyses of LGI, and only two in the SA analyses. However, additional regions pop out in the stratified analysis, which does not make much sense. In addition, given the limited cannabis x group interactions in the primary analyses, the decision to stratify may not be justified. The authors should choose which method they trust most, and conclusions from any should be tempered given that the smallest cell is n=13. Another limitation is the fact that male and female cannabis users appear to differ on a number of important measures, although the authors do not control for any of these in their interaction analyses. Although they report that cannabis users of different sexes were not significantly different in race, fitness, and cannabis characteristics, these analyses are also likely underpowered with n = 13 vs. 23, leading to a false sense of equivalence in these factors. The manuscript also does not present data on nicotine, which could have influenced findings. In the introduction, the authors do not provide strong justification for why they are examining LGI and SA as opposed to other structural brain measures, given that all have significant heterogeneity in this literature. They claim that LGI may be more susceptible to environmental factors such as exogenous drug exposure, although I’m not sure the studies they cite had specific head-to-head comparisons of various structural measures. Moreover, it’s not sure whether surface area also would be more sensitive and why it was chosen in this regard. The authors also do not acknowledge difficulties that have been raised with surface area analyses and local gyrification indices (e.g., Gao et al., Hum Brain Mapp. 2014). The authors discuss findings by describing relationships as “more robust” between aerobic fitness and brain morphometry in women, but discussing robustness of these relationships does not really impart any specific meaning. I think what they mean to say is that these relationships or more strongly positive or negative in one sex versus the other, but this is unclear. In the discussion, the authors do not do an effective job of synthesizing their findings with the prior literature in this area. The fact that their main effects were inconsistent with prior studies despite having similar proportion of males in their sample does not necessarily make sense. Adding a moderator doesn’t change that fact. It is unclear whether increased vs. decreased LGI and SA would be predicted in this sample and why they would be opposite in males vs. females. It would help to think about the actual nature of these interactions and potentially graph them with raw data in a figure alongside the brain maps to see the structure of the data. For example, it is not clear with a less positive relationship between aerobic fitness and surface area might mean. I would recommend that the authors be slightly more cautious with causal language in the manuscript. As they point out in the limitations, causality cannot be determined using the design of this study. However, there are many references to cannabis “effects” or interventions to potentially ameliorate cannabis effects, which undermine their point about causality. I’m not clear why the code and data are not available for this study.

Minor:

On page 2 line 66, the authors discuss a previous study but are not specific enough about brain regions. It would be helpful to have some sort of flow chart for participants who were excluded, the total number of participants screamed, any measures of quality control exclusion, and so forth. In table 2, the authors should specify whether the P value is a uncorrected P or a CWP. The second sentence of the discussion is a run on. In the discussion, the authors reference a meta-analysis on cognitive functioning that covers older adults, which is less relevant here.  There are meta-analyses on adolescents and adults that are more appropriate.